# Recent Progress in Gas Sensor Based on Nanomaterials

**DOI:** 10.3390/mi13060919

**Published:** 2022-06-10

**Authors:** Danyang Lun, Ke Xu

**Affiliations:** School of Electrical & Control Engineering, Shenyang Jianzhu University, Shenyang 110168, China; ldysjzu@163.com

**Keywords:** gas sensor, nanomaterials, various dimensions

## Abstract

Nanomaterials-based gas sensors have great potential for substance detection. This paper first outlines the research of gas sensors composed of various dimensional nanomaterials. Secondly, nanomaterials may become the development direction of a new generation of gas sensors due to their high sensing efficiency, good detection capability and high sensitivity. Through their excellent characteristics, gas sensors also show high responsiveness and sensing ability, which also plays an increasingly important role in the field of electronic skin. We also reviewed the physical sensors formed from nanomaterials in terms of the methods used, the characteristics of each type of sensor, and the advantages and contributions of each study. According to the different kinds of signals they sense, we especially reviewed research on gas sensors composed of different nanomaterials. We also reviewed the different mechanisms, research processes, and advantages of the different ways of constituting gas sensors after sensing signals. According to the techniques used in each study, we reviewed the differences and advantages between traditional and modern methods in detail. We compared and analyzed the main characteristics of gas sensors with various dimensions of nanomaterials. Finally, we summarized and proposed the development direction of gas sensors based on various dimensions of nanomaterials.

## 1. Introduction

The twenty-first century is an era of electronic information technology, and the high development of electronic technology has completely changed human society’s way of life. At the same time, the application of high-efficiency gas sensors in various fields, such as substance detection, is extremely critical now and in the future. However, certain problems are being faced in the development of gas sensors. At present, the quality and working efficiency of gas sensors are relatively low, and there are problems such as the aging process and the unreasonable process. To produce efficient and stable gas sensors, it is essential to select quality materials. Nanomaterials make full use of their far lower scale and excellent characteristics than regular devices and ideally become the core of gas sensor manufacturing materials. Nano-sensors have advanced analysis to the atomic scale, which broadens the application fields of sensors and promotes their production level.

After introducing nanotechnology into the field of gas sensors, it has significantly enhanced the selectivity, improved the sensitivity, reduced the working temperature, and improved the detection performance of gas sensors, as well as promoted a new type of gas sensor. The sensor composed of nanomaterials plays a significant role in the development of sensors. Through Chen Ming’s research, it enhanced the gas sensing properties of carbon nanotube films through electrostatic self-assembly [1]. The recording sensitivity of carbon nanotubes-based gas sensors to nitrogen dioxide at room temperature is 1.97 times higher than that of random equipment of the same dimension, which is due to the use of the specific surface area of a carbon nanotube network, and it dramatically enhances the sensing performance. Ma Defu’s studies demonstrate a visible-light-driven room temperature gas sensor made of novel carbide nanocrystals [2]. The fluorescence emission of carbide nanocrystals has been attributed to light-driven sensing. The device also exhibits good selectivity and stability. Carbides have high adsorption energy in regards to specific gas molecules and low adsorption energy of other gases, which leads to the detection of certain gases.

This overview reviews the current development of gas sensors and introduces new gas sensors based on various nanomaterials, including their composition, types, principles, characteristics and corresponding applications. Nanotechnology has created great potential for manufacturing high sensitivity, low-cost and low power consumption gas sensors. It mainly introduces gas sensors based on various dimension nanomaterials. In each part, it has the methods we use, the characteristics of various categories and the contribution of each study. Herein, we will discuss how to solve the existing problems and combine their elements and applications to compare the main advantages and disadvantages of gas sensors with various dimensions of nanomaterials. Finally, we give a brief summary on nanomaterials with different dimensions in Table 1 and summarize the development direction of gas sensors based on various dimensions of nanomaterials.

## 2. Gas Sensors Based on Zero Dimensional Nanomaterials

### 2.1. Gas Sensors Based on Carbon Dots

Carbon dots have excellent optical properties, good water solubility, low toxicity, environmental friendliness, are a comprehensive source of raw materials and are low-cost with good biocompatibility [3]. As one of the zero-dimensional carbonaceous nanomaterials, carbon dots have many peculiar properties such as quantum size, abundant edges, functional groups, high conductivity and so on in physical chemistry, playing an essential role in the development of the nano-field, which is beneficial to the improvement of gas sensing performance, making it the perfect material for a gas sensor.

Jing Hu synthesized reduced graphene–oxide–carbon dots (rGO-CDs) hybrid materials via green one-pot method to situ generate ultra-small-sized surfaces on their surfaces while reducing GO [4]. The introduction of CDs significantly improves the gas sensing performance of rGO. Composite structures can detect extremely low NO_2_ concentrations at room temperature. Prepared rGO–CDs exhibited high sensitivity and good selectivity for NO_2_ at room temperature upon exposure. It attributed the improvement of rGO–CDs gas sensing performance to the increase of surface hole density rGO surface, which has few defects regarding residual nitrogen due to the introduction of CDs and the formation of an all-carbon nano heterojunction, which itself significantly promotes charge transfer and exhibits high sensitivity, high stability and high repeatability of an NO_2_ gas sensor based on nanomaterials.

This fabrication method forms heterojunctions and a small amount of nitrogen doping in full-carbon nanoscale rGO–CDs, which effectively promotes charge transfer and perfectly demonstrates its good gas sensing performance. In observing the sensing performance of the sensor in the detection rGO-CDs, the sensor has the best performance. It exposed the sensor to air, and the adsorbed oxygen molecules captured electrons from the semiconductor, thereafter obtaining the chemical adsorbed oxygen species. This process resulted in the formation of a depletion layer on the rGO–CDs surface. There are grain boundaries at the same time in many active sites in rGO–CDs and the composite contact gas with NO_2_, and it used the whole carbon nanoscale for the selectivity of NO_2_ heterojunction molecules. The nano-heterostructures and a small amount of n doping in carbon dots significantly promote the charge transfer in the depletion layer of nanomaterials. Because NO_2_ has lone pair electrons and the interface electron exchange and gas will produce a very intense interaction, the sensor resistance changes dramatically. All of these factors affect the electrical characteristics of the system and significantly improve the sensing performance of the sensor.

Cheng Ming studies gas sensors based on carbon dots. He used simple environmentally friendly hydrothermal methods in the In_2_O_3_ nanosphere to change the situation of a delamination situation to improve gas sensing performance. He also used a simple one-step hydrothermal method combined with subsequent annealing processes to prepare uniform and hierarchical In_2_O_3_/carbon-like nanospheres. As validation, it fabricated gas sensors based on In_2_O_3_/carbon dots and investigated their gas sensing properties.

This method makes use of the excellent properties of In_2_O_3_ to make gas sensors. When exposed to the air, oxygen molecules will be adsorbed on the surface of In_2_O_3_. These adsorbed oxygen molecules will be ionized by trapping electrons in the In_2_O_3_ conduction band to form chemically adsorbed oxygen species. The electron depletion layer will form indium oxide near the surface, which will reduce the electron concentration and increase the resistance [5]. When exposed to oxidized gases, the construction of nitrogen dioxide molecules that capture electrons from In_2_O_3_ to NO_2_- increases resistance after combining with the carbon dots, the successful electron transfer, the formation of the heterostructure of In_2_O_3_ and the carbon dots. The shielding effect is due to the introduction of carbon dots and the increased surface adsorption oxygen of the introduction of carbon dots. Carbon dots have several purposes, such as an active surface effect, and they show excellent sensing performance for NO_2_ and other gases.

Compared to traditional gas sensors, In_2_O_3_/carbon point gas sensors have apparent advantages. They form heterojunctions at the interface between the In_2_O_3_ and the carbon dots; electrons flow from the In_2_O_3_ to the carbon dots until the electrical properties are equal; the electron depletion layer widens as the band bends, eventually leading to an increase in resistance. The introduction of carbon dots will produce a shielding effect to reduce the effective nuclear charge. Therefore, the released electrons will be absorbed by nitrogen dioxide molecules, further improving the sensing performance [6]. The percentage of surface adsorption oxygen increases after the introduction of carbon dots. Since the surface adsorption oxygen has high activity, the increase of surface adsorption oxygen percentage plays a vital role in improving the sensing ability. The rich crystal structure can provide more free electron adsorbed oxygen for atoms to react with NO_2_ gas, promoting the sensing performance. The surface of carbon dots has ultra-high stable chemical activity, which is beneficial to the adsorption of NO_2_ gas and oxygen so that the reaction can accelerate gas sensing. Because of its unique structure and characteristics, the gas sensor can significantly improve sensing performance.

Ziyang Yu studied the synthesis of ZnO and carbon dots (CDs) via the hydrothermal process [7]. ZnO/CDs composites were prepared by doping the CDs into the ZnO via the grinding method. X surface area of zinc oxide adsorbed gas can be provided via optical sheet diffraction and scanning electron microscope analysis. The ZnO/CDs composite has a high gas sensitivity response. The gas sensitivity test of the ZnO/CDs composite shows that the sensor has a high NO response. The reaction rate of ZnO/CDs composites to NO is much higher than that of traditional methods, and the active functional groups provided by CDs have a significant effect on the NO.

The most significant difference is that ZnO/CDs composites have enhanced gas response, and the doping CDs have an essential effect on NO. The spontaneous formation of free radicals is a distinctive feature of NO gases. The study on carbon dots introduces the surface of carbon dots with active functional groups, capturing free electrons in the ZnO conduction band in ZnO/CDs composites. When the concentration of the carrier decreases, the conductivity of the material decreases either. The measured resistance value increases, which shows that the gas sensing response of the ZnO/CDs composite is improved and enhanced.

The presence of carbon dots transforms the gas adsorption reaction into the solid-phase contact reaction in the composite, and the electron transfer in the solid-phase contact reaction is more likely to occur. Materials with a porous micromorphology have a large surface area; the larger surface of the material can absorb more oxygen molecules and test the target gas molecules, and more surface contact will occur at this time, resulting in better gas sensing reaction. The microsphere morphology of sheet assemblies with a large specific surface area in ZnO/CDs composites provides more contact with no gases [8]. The numerous active functional groups doped with CDs provide more non-gas-sensing reaction sites, so this method can increase the gas sensitivity of the material and improve the efficiency of the gas sensor.

### 2.2. Gas Sensors Based on Nanoclusters

In recent years, nanoclusters have become a new material in the field of nano-research. More and more researchers have paid attention to them, including, specifically, gold nanoclusters, as they are considered the most typical representative of metal nanoclusters, having more engagement. Gold nanoclusters are molecular level aggregates with fluorescence properties prepared from organic molecules as templates. Their size is similar to a fermi wavelength and can produce specific energy level separation. Therefore, it emitted fluorescence under excitation at a specific wavelength. Compared to traditional fluorescent materials, such as organic fluorescent dyes and nanoparticles, gold nanoclusters have become excellent materials for gas sensors because of their simple preparation methods and unique physicochemical properties. Nanoclusters have developed well in the research and manufacture of gas sensors because of their rich characteristics, such as light stability, excellent biocompatibility, light induced flourescence and outstanding sensing performance.

Hossain Khan studied a highly sensitive and selective nitrogen dioxide detection method [9], which itself functionalized Gallium Nitride (GaN) submicron wires with titanium dioxide (TiO_2_). The nanoclusters fabricated dual-terminal gas/TiO_2_ sensor devices using a top-down approach. Gas sensing makes it possible for the sensor to work at room temperature. After the study, it was found that the sensor had high selectivity to NO_2_ and can resist other interfering substances. The sensor device had good long-term performance stability at room temperature and humidity, and is relatively stable and reliable in various climates.

This study uses metal oxide nanocluster functionalized GaN sensors to realize the sensing of NO_2_ molecules. Under UV irradiation, metal oxide nanocluster photolysis water absorption and water in the GaN create oxygen-producing surface defect active sites and electron-hole pair frameworks; target analytes undergo chemisorption at these active sites; adsorption molecules dynamically capture and de-capture charge carriers at these active sites for GaN potential modifications of the main chains, leading to modulation of sensor currents, proportional to analyte concentrations [10]. Oxygen molecules are chemisorbed on Ti^3+^ vacancies on the TiO_2_ surface to obtain negative charges. Meanwhile, molecular adsorption or dissociation adsorption occurs on the surface of water molecules, and TiO_2_ cluster surfaces produce oh substances at the Ti^3+^ defect sites, which also have many advantages as a sensor. When the energy is higher than the bandgap energy of GaN and TiO_2_, it activates electron-hole pairs in GaN and TiO_2_ clusters under UV excitation [11]. The carrier lifetime increases in the GaN submicron line due to the rise of photocurrent increases due to the bending of the surface energy band photogenic pores to the GaN surface. Chemisorbed oxygen and water are received TiO_2_ molecules on nanoclusters and desorbed [12]. It adsorbs Nitrogen dioxide directly on these newly generated sites due to the high affinity of molecules. Some NO_2_ molecules interact with and are adsorbed on the surface chemically adsorbed oxygen. TiO_2_ nanoclusters and NO_2_ molecules increase the depletion region width inside the GaN, thus reducing the sensitivity current of the sensor. The desorption of light-induced oxygen and subsequent charge transfer TiO_2_ nanoclusters and NO_2_ molecules to regulate the depletion region width within the GaN, thus contributing to high-performance NO_2_ gas sensing.

Mingyuan Wang studied a gas nanosensor system combining silver nanoclusters with phosphorene [13]. The Ag_N_ nanoclusters (1 ≤ N ≤ 13) can effectively reduce the degradation of phosphorene and hypophosphorous in the catalyst and exhibit various structures. Exposure to other active adsorbents can play a good adsorption role, which significantly improves the selectivity and sensitivity of the system to adsorbed molecules. Because of the participation of valence electrons, the modification of silver atoms from electron orbit can improve the sensitivity of phosphorene. It can also regulate the charge distribution between atoms to adsorb molecules and phosphorene. When the gas flows, the work function of the molecules adsorbed on Ag_1_ phosphorene changes significantly, the adsorption amount of NO_2_ molecules increases significantly, and NO_2_ adsorption requires higher bias voltage than that of Ag_1_ phosphorene. Then, the sensing of NO_2_ gas is achieved.

This study proposes a silver-trimmed phosphorene composite system for gas sensing, which can prevent silver aggregation nanoclusters and reduce the degradation and passivation of phosphorene by metals through stronger interactions. It improved the selectivity and sensitivity by adjusting the adsorption energy and temperature [14]. In this study, the reagent has good selectivity and stability. The sensitivity of four molecules to phosphorene can significantly improve the modification of silver nanoclusters via surface modification. Single Ag decoration can improve the adsorption performance of NO_2_ molecules, enhance the sensitivity and selectivity of NO_2_ molecules and make it an ideal material for gas sensors.

The excellence of this experiment is that phosphorene is a suitable scattering substrate. Silver nanoclusters are efficiently sensed by increasing the binding of silver phosphorene and weakening the silver–silver bond. Single Ag and Ag_1_ phosphorene show that Ag atoms cannot only increase the adsorption energy but also increase the energy difference, which means that the phosphorene modified by single silver can improve the sensitivity of the four gas molecules [15]. NO_2_ adsorption requires higher bias voltage than single silver-modified phosphorene. With the increase of the number of Ag nanoclusters, the stability of Ag_N_ nanoclusters to the adsorption energy of gas molecules is enhanced, which provides a variety of ways for the selective adsorption of gas molecules. Silver decoration can create new synergistic effects, improve efficiency by adjusting adsorption strength and temperature, and then enhance the sensitivity and selection characteristics of the phosphorene surface as a gas sensing element.

In YoungMin Byoun’s research, they synthesized heterostructured nanomaterials composed of p-type TeO_2_ NWs and discrete n-type ZnO nanoclusters to detect NO_2_ gas molecules [16]. These nanomaterials are synthesized via thermal evaporation and atomic layer deposition, and then the ability of NO_2_ in terms of operating temperature, NO_2_ response and selectivity was systematically investigated [17]. By forming discrete n–ZnO nanocrystals, it enhanced the reaction of the p-TeO_2_ nanocrystalline sensor to NO_2_ significantly. The synthesized sensors also show good NO_2_ selectivity compared with SO_2_, C_2_H_5_OH and other interfering gases, forming discrete n–ZnO nanocrystals to improve the sensing ability of gas sensors to NO_2_ significantly.

This study demonstrates an excellent reaction and results. ZnO-TeO_2_ heterostructures exhibit good NO_2_ gas sensing performance by electron sensitization. The heterojunction generated by n–ZnO functionalization can give the p–TeO_2_ nanowires better resistance; besides, heterojunction nanowires are beneficial to the detection of oxidizing gases, and when p–TeO_2_ contact n–ZnO, the flow of electrons from n-ZnO to p–TeO_2_—transferred to the n–ZnO to balance the Fermi level—results in a barrier with band bending [18]. As a result, the relative change of hal volume caused by adsorption and desorption of oxidant is more significant than that of reductive gas. Compared to SO_2_ molecules, NO_2_ molecules are more readily adsorbed on the surface ZnO–TeO_2_ heterostructure nanocrystals. As a result, the response to the NO_2_ is significantly enhanced.

The advantage of this study is the synthesis of heterostructures p–TeO_2_, the continuous functionalization of n–ZnO nanocrystals using thermal evaporation and ALD processes. It characterized and tested the synthetic products and systematically studied the SEM, XPS and NO_2_ of the models. Fabricated sensors show significantly improved NO_2_ sensing capabilities far better than the original sensors. Specifically, n–ZnO nanocrystals have a positive influence on the NO_2_ response. The n–ZnO nanocrystals are functionalized on the surface of the p–TeO_2_ nanocrystals, thus demonstrating good sensing performance for the gas.

### 2.3. Gas Sensors Based on Metal Nanoparticles

With a deepening of the research of nanomaterials, the huge application potential of gold nanoparticles in the field of gas detection technology has been widely recognized. Nano gas probes and their corresponding detection technology have been highly valued. In recent years, hybrid systems of gas molecules and spherical gold nanoparticles have been widely used in various biological analyses and have achieved encouraging results. Due to the characteristics of easy preparation, easy biochemical modification, high density and high dielectric constant, it used gold nanoparticles to prepare gas sensors to detect specific gases.

Sh.Nasresfahani investigated the effects of gold nanoparticles on the performance of CO gas sensing sensors [19]. It comprehensively studied room temperature properties of polyaniline and prepared Au/PAni nanocomposites. Also, it modified the fiber surface due to electrostatic interaction and hydrogen bonding PAni via emission scanning electron microscopy and x-ray spectroscopy analysis. Then, the study analyzed the gas sensitivity of each sensor to various carbon monoxide gases in a concise range. Experiments show that the Au/PAni sensor has high response and low noise, a very short response time, a wide dynamic range and good stability. The catalytic performance of gold nanoparticles determines their selectivity and the good sensing ability of the sensor.

The difference between this study and the traditional method is that it enhanced the sensitivity of p semiconductor PAni to CO gas after introducing au nanoparticles. The positive direction on the carbon atom transfers to the nitrogen on the amine, which increases the amount of the positive charge, so the conductivity of polyaniline increases. When it introduced gold nanoparticles into polyaniline, the Au–NPs can interact with CO molecules and transfer positive charges to polyaniline, which significantly improves the sensitivity of the sensor [20]. It prepared Au/PAni nano composites by ultrasonic mixing under appropriate conditions. The physical mixing of the two components resulted in negatively charged metal nanoparticles deposited on the positively charged polyaniline surface. Since the high surface energy of gold nanoparticles creates adsorption sites for CO gas molecules, the prepared sensors exhibit good gas sensing properties for various concentrations CO at room temperature.

Do Wan Kim proposed a highly sensitive and rapidly responsive nitrogen dioxide gas sensor based on gold nanoparticles modified zinc oxide nanowires. On the surface of zinc oxide nanowires, it modified gold nanoparticles on its surface by electrostatic force. The models without the aptes layers exhibit high NO_2_ gas sensitivity due to the local surface plasmon resonance (LSPR), and, incredibly, the increase of the NO_2_ gas response and response time was three-fold. Compared to the unmodified ZnO nanowires, the time was reduced by 80%. The presence of aptes layer improves the attachment of gold nanoparticles, and the LSPR effect can significantly improve the efficiency of gas sensors.

Figure 1 illustrates the machine-made strengthened NO_2_ gas response of Au–ZnO and Au–ZnO/APTES through diagrams, especially the absorption and suction mechanisms in the dark and under green clearing luminary. We discussed two typical adsorption avenues here: the immediate chemisorption of NO_2_ gas onto the ZnO surface by catching the electrons of the ZnO surface itself and the removement of the NO ions from the Au NPs where NO_2_ gas catches the electrons of the Au NPs. The suction of the NO ions should only emerge on the ZnO surface. As shown in Figure 1a, the Au NPs adherence should enhance the NO_2_ gas adsorption under green lighting by engendering plasmon-mediated hot electrons from the Au NPs. In detail, both the afflux of hot electrons and the light stimulates electrons from the defect levels broaden the conducting channel of the ZnO NW under green light. Then, the NO_2_ gas adsorption on the ZnO surface is extended by the increasing number of electrons, extending the depletion region by capturing the electrons. Then, the electric channel of the ZnO NW becomes narrower, as revealed by the red hollow cylinders in Figure 1a. As a result, Au–ZnO achieved the maximum improvement ratio and NO_2_ gas response. Even though the suction of the adsorbed NO_2_ and O_2_ ions was also enhanced by the generation of holes in the ZnO NW under green lighting, the hotelectrons-associated gas adsorption procedure appeared to be improved.

On the other side, the APTES layer on the ZnO NW surface exerts an influence in the NO_2_ gas absorption of Au–ZnO/APTES. The APTES layer hindered the NO_2_ gas adsorption onto the ZnO NW, as certified by the PC decay results revealed in Figure 1a,b and testified by the comparatively thin arrows of NO_2_ in Figure 1b. In contrast, once the NO_2_ gas was adsorbed onto the ZnO NW surface of Au–ZnO/APTES, its suction was comparatively less hindered by the APTES layer compared to the absorption procedure. Au–ZnO/APTES showed a broadened electric channel of the ZnO NW under green lighting due to the afflux of hot electrons and the light stimulates electrons, similar to that of Au–ZnO. However, since the NO_2_ gas adsorption on the ZnO NW surface was not abundantly expedited due to the hindering of the APTES layer, the narrowing in the electric channel of the ZnO NW was smaller compared to the Au–ZnO case. This could explain the comparatively small NO_2_ gas response of Au–ZnO/APTES under green lighting with respect to Au–ZnO despite the LSPR effect of the Au NPs. Furthermore, for all three samples, both the red and green lighting obviously expedited the NO_2_ gas suction procedure more than the adsorption one, except for the green lighting of Au–ZnO, where the LSPR effect was outstanding. Consequently, the NO_2_ gas sensing mechanism of the samples could be directly proofed with the unity of the hot electron generation from the Au NPs via LSPR and the gas adsorption reduction by the APTES layer.

In the research of a gas sensor, there is a unique treatment. Plasma mediated enhanced the room temperature NO_2_ gas sensing properties of gold, NPS modified ZnO nanowires, and proposed potential mechanisms for improved LSPR effects. The charge transfer resulting from the LSPR effect in Au–ZnO was characterized via light-irradiated (red and green, 650 and 532 nm, respectively) KPFM measurements. Compared with pristine ZnO nanowires, they have higher gas sensing performance, uniquely faster response time. Gas-sensitive properties of this hot electron increase sharply because it captured the hot electron by NO_2_ gas easily and accelerates its chemisorption on the surface of zinc oxide [21]. Stimulated by hot electrons, it replaced the adsorption process of NO_2_ gas by light-excited holes to understand the absorption process [22]. The LSPR effect of hot electrons can improve the response to NO_2_ gases and significantly accelerate the response time of ZnO films. The performance of gas sensors can be significantly improved by using the LSPR effect.

Pu Li proposed in this study to fabricate microstructured gold nanoparticles functionalized gas sensors that are assembled and deposited between two electrodes. In response to volatile organic compounds, it determined the change of conductivity by interparticle properties such as a dielectric constant. The micro sensor shows the reaction of seven target analytes to o-xylene [23]. This micro-sensor exhibits a larger response to o-xylene than conventional sensors, improving sensitivity and shortening the response time to other volatile organic compounds because the larger surface volume results in better sensing performance than surface roughness and device miniaturization [24]. The gas sensor, composed of gold nanoparticles, produces an obvious reaction in a short reaction time.

An SEM image shows the surface morphology film of the sensor, which indicates that the microsensor has large surface roughness and surface volume ratio. The higher response is due to the enhanced surface effect, and the larger surface volume ratio can shorten the precipitation time of the gas. The miniaturization of gas sensors brings higher response speed and shorter response time. We compared the response time of various micro-sensors. For gas sensors based on gold nanoparticles, the response time of the VOC test is much shorter than that of other volatile gases. Compared to the laser-made micro-sensor by writing technology, the micro-sensor shows the excellent response to gas.

Electron transport between particles in this study is a unique method and there is no chemical reaction between nanoparticles and gas molecules. Nanoparticles are exposed to analytes, and the VOCs of the adsorption sensor surface can cause changes in physical parameters [25]. When the polar gas molecules diffuse in the air, the sensor, composed of gold nanoparticles and their binding nanoparticles, expands the distance particles between the particles. The diffusion analyte changes the permittivity constant and the interparticle distance. Using laser writing techniques, gold nanoparticles were successfully self-assembled at two electrodes. Gas sensors exhibit the selectivity of gold nanoparticles in 7 analytes [26]. Due to the excellent performance of the rough surface and high response and short response with the miniaturization of the device, the gas sensor, based on gold nanoparticles, shows a very high response-ability to gas matter.

Ahmad I. Ayesh investigated PbS nanocrystals, and this substance has recently shown room temperature sensing capability for specific gases (CH_4_) [27]. Gold is another common noble metal used as an additive to improve device performance. The incorporation of gold nanoparticles (NPs) can improve methane sensing properties. PbS-NCs shows that adding appropriate amount of Au–NPs can improve PbS-NCs methane sensing characteristics, and studied and analyzed the conductivity, sensor response and sensor speed.

In this study, it studied the effect of gold nanoparticles on PbS electrical properties through various properties. Introducing Au–NPs into the PbS-NCs is feasible to reduce their electrical conductivity. The reaction of the Au–NPs sensor modified on the PbS-NCs surface improves its speed. The main reason is that Au–NPs can produce more oxygen adsorbed on the PbS–ncs surface, and methane molecules find that more adsorbed oxygen will affect the PbS–ncs surface [28]. In similar cases, for the same concentration of methane, it involved more oxygen ions by methane. This not only makes the sensor more sensitive but also more efficient than in the traditional way. The enhancement of gold to oxygen adsorption can improve the properties of the sensor by competing methane molecules and adsorbed oxygen interaction between gold and gold nuclei. With low methane concentrations, when we added PbS-NCs as more gold nuclear power sources changes, PbS–NCs occurred, and it adsorbed more oxygen by methane molecules and more essential conduction. At high methane concentration, there will be competition between gold nuclear power sources and interaction between CH_4_ molecules and adsorbed oxygen, which will produce a better gas response. The effect of various gold content on PbS–NCs decoration affect NPs conductivity and the effect of methane sensing significantly improves its performance. The appropriate amount of gold nanoparticles enhanced the efficiency of the reaction and forms a more efficient gas sensor. (This section closely follows the topic to discuss. The knowledge of nano aspects is well used to analyze and solve the sensor problems. In this part, by reading a large number of data about gas sensors composed of nano materials, the experimental processes of different methods are analyzed in detail, and the analyzed data are compared. The advantages and disadvantages of different kinds of nano sensors are obtained, which shows the diversity of nano sensors.)

## 3. Gas Sensor Based on One Dimensional Nanomaterials

### 3.1. Gas Sensors Based on Nanowire

The continuous progress of global industrialization not only improves production and living standards but also destroys the environment in varying degrees [29]. With the increasing diversity and complexity of harmful gas components in the background in recent years, human health and production safety are in crisis, and people’s awareness of self-protection is improved. Therefore, it is necessary to realize the real-time monitoring of toxic and harmful gases in the environment. Because of its advantages of high sensitivity, easy preparation and low cost, nanowire materials have developed well and play an essential role in the market, and the device performance has been gradually improved nanowire material as the carrier. By regulating its structure and morphology and exploring the synergistic recombination with other semiconductor metal oxides, we try to construct the correlation between material characteristics, gas sensing performance and sensing mechanism, and then improve the gas sensing performance of the sensor composed by it, which lays a foundation for further development in the future.

Zhicheng Cai synthesises SnO_2_ nanowires modified by Pd nanoparticles to prepare highly selective and sensitive nanowire hydrogen sensors [30]. It prepared the SnO_2_ nanowires by steam–liquid–solid process and modified the Pd nanoparticles by UV with a PdCl_2_ solution to improve the hydrogen sensing performance of the SnO_2_ nanowires. Pd nanoparticle-modified SnO_2_ nanowires have good electrochemical performance, and various hydrogen-sensitive responses increase with an increasing number of Pd nanoparticles. Furthermore, the selectivity of this nanowire-based sensor also increases the nanoparticle with increasing Pd. SnO_2_ and the sensing response of nanowires to several gases is similar, as they enhanced the hydrogen sensing response to other gases after various palladium nanoparticle modifications significantly.

Another special feature of this research is that when SnO_2_ nanowires are exposed to the air, the oxygen in the air is adsorbed on the surface of the nanowires due to the attraction of static electricity. It converted the adsorbed oxygen into oxygen ions to adsorb electrons on the surface of SnO_2_ nanowires, SnO_2_ the surface depletion layer of nanowires expands and the resistance increases. When it exposed such nanowires to hydrogen, it adsorbed hydrogen on the surface of SnO_2_ nanowires. Hydrogen reacts with oxygen ions adsorbed on the SnO_2_ surface, and hydrogen is converted into H_2_O gas. Through this reaction, electrons absorbed by oxygen ions return to the SnO_2_ nanowires, SnO_2_ the carrier concentration, surface width and resistance of the nanowires return to the initial state. In this case, the palladium adsorbed nanoparticles, after the reaction with palladium nanoparticles, the electronic band structure changed and the initial dl was formed on the surface of SnO_2_ nanowires when it exposed Pd nanoparticle-modified SnO_2_ nanowires to air, it adsorbed oxygen by electrons of Pd nanoparticles. SnO_2_ the surface of nanowires, it significantly reduced the resistance of such nanowires in the hydrogen environment due to the SnO_2_ use of electrons as electrical carriers. After modifying Pd nanoparticles, the behavior of this nanowire in exposure to air and hydrogen and the change of gas sensing characteristics were more pronounced.

Tzu-Feng Weng used the vapor–liquid–solid growth method to grow high-density β-Ga_2_O_3_ single crystal nanowires on silicon substrates, and studied the room temperature CO gas sensor of pure nanowires and gold-modified nanowires using multi-network arrays and single nanowire devices [31]. It studied the synthesized nanowires by field emission scanning electron microscopy. It fabricated single nanowire gas sensors by focusing ion beam technique. A single nanowire RT-CO gas sensing sensor using the proposed Au changed the β-Ga_2_O_3_ nanowire to achieve remarkable sensitivity to CO gas at room temperature. It also compared the sensing characteristics β-Ga_2_O_3_ RT-CO gas multi-network Au modified nanowires and single Au modified nanowires.

This study, due to the superior and stable RT gas sensing properties of gold, analyzed the effect of various gas concentrations on the performance of β–Ga_2_O_3_ nanowire devices after modification. Various gas sensors have various measurement results in CO gas concentration. With the decrease of gas concentration, the response time and recovery time gradually decrease and increase. The activation energy of oxygen atoms pre-absorbed on the surface of gold nanoparticles adsorbed upon CO molecules decrease [32]. The length of response and recovery time depends on the availability of a large amount of oxygen trapped on the surface of the sample. If the number of oxygen vacancies increases, the pressure on the surface of the number of trapped oxygen molecules increases, which in turn enhances the efficiency of the sensor. It operated the CO-gas sensor by adsorption, the resistance decreases when the reducing agent reacts on the surface of the material. After it chemisorbed and absorbed the oxygen in the semiconductor by the reduced gas, the result is a free electron in the form of increased conductivity. The gas sensing structure diagram is shown in Figure 2. Small fragments of the oxygen ion monolayer were absorded and formed additional deoxyribonucleic acids near the ion surface [33]. This is also the reason for the excellent gas sensing performance of Au modified β-Ga_2_O_3_.

J.Y. Lin studied an SnO_2_ nanowire hall-effect gas sensor for hydrogen detection [34]. It prepared SnO_2_ nanowires on stainless steel mesh by horizontal electric furnace, and it analyzed the crystal structure, morphology and electron binding energy of SnO_2_ nanowires by XRD, XPS. Gas response to H_2_ at various operating temperatures and H_2_ concentrations. It investigated a response mechanism of a SnO_2_-based hall-effect gas sensor. Hall effect gas sensors based on SnO_2_ nanowires have super-high response and efficiency in preparing low-cost and high-performance gas sensors.

Compared to the experimental results of pure SnO_2_ gas sensors, the response characteristics of hall-effect sensors are better. SnO_2_ nanorods evaporation was synthesized via thermal method, which has a high response to H_2_. The sensing mechanism of target gas and hall effect principles with the surface reaction of hydrogen and oxygen was adsorbed on metal surface SnO_2_ nanowires. It absorbed the electrons by atoms to adsorb oxygen molecules from the conduction band of electrons to form SnO_2_ nanowires and oxygen ions on the surface, which leads to the decrease of electron density on the surface of SnO_2_ nanowires [35]. The carrier concentration increased with time, and it exposed the gas sensor to hydrogen. With the growth of hydrogen concentration, the carrier concentration increased further. When it introduced hydrogen, hydrogen reacted with adsorbed oxygen. The absolute value of the hall coefficient increased with the increase of H_2_ concentration, and the decrease of hall voltage decreased with the increase of H_2_ concentration. Good sensing performance was therefore shown in response to a specific gas.

Chandan Samanta studied an NO gas sensor based on ZnO/Si nanowire heterojunction arrays operating at room temperature with extremely high response [36]. The sensor is highly selective for gas-free and confined gases due to the disturbance caused by moisture. Using economical and efficient chemical treatment compatible with wafer-level treatment, the sensor prepared vertically oriented nano-silica arrays by chemical corrosion and zinc oxide deposition, and prepared nanostructures by chemical solution deposition and spin-coating. The formation of heterostructures leads to a synergistic effect in which the sensing response is larger than the sum of individual components, zinc oxide and silicon nanowires. When it combined the n-ZnO nanostructures with the p-sinw interface, the reaction is powerful, which leads to a good sensing response to the gas.

In this way, it made gas sensors, and the response performance of the equipment increases with the decrease of moisture in the environment. It shows the variation of gas response with various values of temperature. With the increase of temperature, the reaction increases ZnO/Si nanowire devices respond to NO gases. Selectivity to a particular compound is a unique feature of the gas sensor. Checking the selectivity for NO gas, ZnO/p-Si nanowire sensor tested for contact with various types of gas. It showed the selectivity when exposed to various kinds of gases, and the gas sensors all had a good response [37]. The gas response of the device, such as nitrogen dioxide, was tested with other oxidants, the nanowire sensor has high selectivity to NO gas. ZnO/p-Si nanowire sensors also generated gas-sensitive responses to various gas lasers. ZnO/Si-NW heterostructures enhanced the performance in semiconductor materials, reflecting ZnO/Si role of nanowires. The room temperature conductive response nanostructured films and p-Si nanowires ZnO by measuring the heterostructure of NO gas showed the high response and sensing ability of the sensor to the gas.

Waldir Avansi Jr. studied the gas sensing properties of semiconductor nanomaterials with various energy bands, the chemical resistance sensing ability of titanium dioxide nanoparticles and the V_2_O_5_ nanowires obtained by hydrothermal treatment of metal peroxide complexes [38]. The formation of V_2_O_5_/TiO_2_ heterostructures was studied, characterized by X-ray diffraction (TEM) and X-ray photoelectron spectroscopy (XPS) measurement. This research also proposes an effective method for preparing one-dimensional vanadium pentoxide/titanium dioxide with good detection range and ozone sensing properties that are significantly related to repeatability and selectivity.

The difference in this study is that both V_2_O_5_ and TiO_2_ are n-type semiconductors with various structural electronic properties, such as electron affinity. When equilibrium occurs between semiconductors, electrons transfer from TiO_2_ to the low-energy conduction band V_2_O_5_ the semiconductor until their fermi levels become equal. This arrangement leads to the presence of more electron conduction bands in the TiO_2_. It attributed the ozone response of the heterostructure to the effect of the existence of carbon impurities on the active sites. For V_2_O_5_/TiO_2_, the heterostructure presents various kinetic curves, and it observed the response of the sensor, indicating that the formation of the heterojunction produces additional active sites. It connected the sensor response enhancement O_3_ gas V_2_O_5_/TiO_2_ heterostructure to the induced effect nanowires and titanium dioxide nanoparticles of vanadium pentoxide effective heterojunction at the two-phase interface, which leads to the change of resistance and then improves the response to the gas [39]. Synergistic effect promotes the chemical adsorption process of ozone. The gas sensor has good gas sensing performance and a strong sensor response to ozone gas.

### 3.2. Gas Sensors Based on Carbon Nanotubes

With the development of nanotechnology, it has created great potential for designing low energy consumption, high sensitivity, low cost and portable sensors. As emerging nanomaterials, carbon nanotubes’ excellent electrical conductivity, high surface area and unique hollow structure make them ideal materials for gas molecule adsorption. The commonly used semiconductor metal oxide gas sensing materials usually need to work typically at a higher temperature, showing semiconductor characteristics [40]. The incorporation of carbon nanotubes affects the gas sensing characteristics of semiconductor oxides, and its composite gas sensing materials show good gas sensing characteristics.

Sunil Kumar studied the chemical detection of toxic gases such as greenhouse gases by gas sensors based on single-walled carbon nanotubes (SWCNTs) to detect NO_2_ gases with higher sensitivity [41]. In this study, the thin film sensor was fabricated on a SiO_2_ substrate and functionalized with polyethylenimine (PEI) respectively. Confirmed that the PEI functionalized SWCNTs showed a high sensitivity for strong electron-absorbing solid. At room temperature, the sensitivity of SWCNTs that PEI functionalized gas sensing elements is nearly 50% higher than that of single-wall carbon nanotube gas sensing elements. The gas sensor shows a repeated response over the entire study concentration range. PEI functionalization improves the performance of single wall carbon nanotube gas sensing elements.

This study used the thermal CVD method to develop resistive SWCNTs-PEI functional gas sensors. PEI coated single-walled carbon nanotubes have higher electron-absorbing NO_2_ adhesion coefficient than untreated single-walled carbon nanotubes. The resistive gas sensor with PEI–SWCNTs coating exhibits high sensing performance and rapid response to NO_2_ at room temperature [42]. Due to the room temperature working characteristics of resistive gas sensors, more apparent results can be obtained in environmental observation. Through the accurate selection of the heat treatment process, it realized the complete recovery of the sensor and the sensitivity of SWCNTs increases with the extension of functionalization time. By controlling the alkalinity and functionalization concentration of single-walled carbon nanotube networks, PEI- SWCNTs sensors can be extended to various fields, and the selection of sensors can be configured according to various chemical environments. All of them show good response to gases [43].

Sukhananazerin Abdulla studied the formation of thin films with good polyaniline arrangements [44]. The high directional ordering of multi-walled carbon nanotubes is enhanced by Langmuir-Blodgett technology to improve the sensing characteristics of ammonia gas. In the process of interfacial assembly, polyaniline-multi-walled carbon nanotubes gradually form ordered small blocks at the gas-water interface and further organize into a globally well-defined oriented monolayer. The PANI@MWCNTs LB films were transferred at 25 °C onto the pre-cleaned gold sputtered double electrodes fabricated on SiO_2_ substrates for gas sensor property analysis. Schematic illustration of the experimental setup used for gas sensing measurements is shown in Figure 3. Direction and p-polyaniline multi-walled carbon nanotubes systematically study the Langmuir film at the air–water interface as the key to an ammonia gas sensor.

The difference in this study is that the surface-functionalized polyaniline-modified multi-walled carbon nanotubes overcome the surface defects caused by the three-dimensional aggregation of carbon nanotubes. Formation stability of dense body single molecular membrane/multilayer structure polyaniline-multiwalled carbon nanotube LB based film. It used highly polyaniline functionalized multi-walled carbon nanotubes with oriented LB films for sensing at room temperature of ammonia gas. Sensitivity to NH_3_ directional electron transport polyaniline multiwalled carbon nanotubes at room temperature compared to random networks [45]. Ultrathin LB membranes allow for fast analyte diffusion of active sensing layer assembly due to adequate molecular regulation. This study resulted in the formation of aligned assembled polyaniline-multi-walled carbon nanotubes, which formed remarkable monolayers at the air–water interface on large areas parallel to the barrier layer dense films. LB membrane-oriented polyaniline multi-walled carbon nanotubes exhibit high-performance gas sensing elements and sense ammonia gas [46].

Florin C. Loghin studied a transparent gas sensor on carbon nanotubes [47]. It deposited the sensing layer and electrode by jet deposited carbon nanotubes. High transmittance transparent sensor electrodes of the two sensing layers have characterized the properties of ammonia and carbon dioxide, and the sensitive reference sensor and transparent sensor for the NH_3_ have shown a good response. In contract, the transparent device has higher sensitivity to carbon dioxide than the reference electrode. The effect electrodes with spacings between continuous data were also investigated with wider spacing in the results of fully carbon nanotubes-based sensors at higher sensitivity due to higher sensing resistance, whereas this effect was not observed in gold electrodes due to their negligible resistance relative to carbon nanotubes. The performance of a transparent sensor is better than that of other sensors, which shows good sensing characteristics of transparent gas.

Throughout this study, a unique point has been that the spacing plays an essential role in high resistance, and this effect will eventually affect the saturation distance of the electrode. Normalized response of the Carbon Nanotubes electrode will be concentrated to one of the Au electrodes if the resistance ratio of the sensing layer to the electrode is large enough. However, this studies a higher electrode distance, which will affect the total resistance of the sensor of the method, and the sensing performance of the translucent carbon dioxide sensor can be significantly improved. At high concentrations, the sensitivity of the sensor is better than that of some all-carbon nanotube gas sensors [48]. This effect may respond to the presence of electrodes in carbon-based sensors against gas molecules. However, they react variously to gases because of the various network density, plus the performance of carbon nanotube electrodes is superior to that of gold-contact carbon dioxide electrodes. At higher concentrations, carbon dioxide can cause a massive change in electrode resistance, resulting in an increasing in distance-insensitive normalized resistance, which then improves gas sensitivity.

Qian Rong used molecular imprinting technology to prepare high-performance acetone gas sensors ALFOMMIPs, studied the functional groups, grain size and surface morphology of synthetic materials, analyzed them through different characterization techniques and examined the gas response of the samples [49]. The CNTs and ALFOMMIPs nanocomposites (CNT/ALFOMMIP) showed higher thermal stability than the reaction of ALFOMMIPs models via an acetone gas sensing test and analysis. The sensor with carbon nanotubes has good gas stability sensing performance. The sensor shows a high response to acetone at various temperatures, and it has the best selectivity and sensing performance for acetone vapor due to molecular imprinting technology.

The porosity of the resistance CNT/ALFOMMIP is related to the number of holes [50]. When we placed CNT/ALFOMMIP gas-sensitive materials in air, chemisorbed oxygen molecules and physically adsorbed on the surface of objects CNT/ALFOMMIP gas-sensitive materials form depletion layers near the sensor surface with ALFOMMIP conductive electrons, reducing the adsorbed O_2_ molecules to various forms of oxygen-containing anions and resulting in reduced electron density and increased sensor resistance [51]. When the acetone vapor in the air is in contact with the CNT/ALFOMMIP gas-sensitive material, the chemically adsorbed oxygen on the surface of the material will evaporate with acetone, producing the extracted electrons released into the CNT/ALFOMMIP conduction band. It significantly increased the resistance of the holes of the carbon nanotubes rapidly transferred to the ALFOMMIPs, CNT/ALFOMMIP sensor. The CNTs have good electrical conductivity, and the resistance of the sensor can be reduced due to the abundant groups such as -COOH or -OH surface CNTs. The electron passes through the percolation effect, which significantly improves the transmission rate. CNT/ALFOMMIP composites have a high response speed and low operating temperature. p-p homojunction is formed via close contact with metal alfomips carbon nanotubes that are prepared, which further improves the gas sensitivity of the sensor to acetone vapor.

Struzzi C investigated the modification of electronic properties on the surface of vertically aligned and randomly distributed films [52]. It used the hydrophobic properties of CNTs in Ar, F_2_ and CF_4_ plasma to optimize these sensing properties. By detecting the stability and responsiveness of fluorinated carbon nanotubes to water, reflecting the sensor sensing characteristics, fluorinated carbon nanotubes were revealed to detect two selected pollutants, such as nitrogen dioxide and ammonia (NO_2_ and NH_3_). By increasing the surface hydrophobic humidity level and the influence of sensing layer geometry on fluorination, resulting in response to reproducibility when used vertically and an enhanced sensor response of CNTs to NH_3_.

The unique feature of this gas sensor is that the fluorination reaction introduces a moderate induction reaction, which reduces the distance between ammonia and the predicted analyte and fluoride, promoting stronger interactions through the formation of hydrogen bonds, due to dipole electrostatic interactions providing enhanced charge transfer with the fluorinated carbon layer and confirming the significant role of CNTs forests in vertical geometry directly exposed to the tip playback of the target gas. The change of humidity concentration in the environment also has a great influence on the performance of nano-pipe network sensors. Fluorination uses CF_4_ plasma, increasing the effect of humidity levels on the response, leading to the generation of an ionic current that promotes the generation of local space charges [53]. The original random CNTs models reacted to ammonia already in the dry state, and there was a hydrogen bond interaction on the surface of natural oxygen in the air, which was beneficial to charge transfer to the fluorinated CNTs sensor. The fluorination CNTs then showed good sensing characteristics for the gas (The topic selection has strong application value literature materials collected detailed, the knowledge learned has solved the problem, correct conclusions, innovative insights, experimental design is reasonable and feasible, can be carried out according to the experimental plan, and achieve the expected results).

### 3.3. Gas Sensors Based on Other One Dimensional Materials

Fei Yang studied a visible-light-driven room temperature gas sensor made of a novel carbon-acetylene nanocrystalline system [54]. It prepared nanocrystals via laser ablation. Scanning electron microscopy and transmission electron microscopy show that they are stacked flakes composed of rod-like crystals. Under illumination at 447 nm, a sensor containing nanocrystals detected NO_2_ molecules with concentrations as low as 2 ppm at room temperature with a response and recovery time of less than 100 s. It attributed the light-driven sensing to the fluorescence emission of carbon acetylene nanocrystals. The device exhibits excellent selectivity and stability. The carbon nanocrystals have high adsorption energy NO_2_ and the molecules have low adsorption energy to other gases, which leads to high NO_2_ detection sensitivity.

The gas sensor fabricated in this way demonstrates the potential of carbon nanocrystals as a gas-sensing material. NO_2_ adsorption is a non-dissociated process, is not determined by thermal energy. Because of the high absorption rate of the NO_2_, the NO_2_ adsorption sites have limited barrier heights and a limited number of photogenerated electrons. This Au load effectively reduces the Au–C adsorption barrier Schottky junction, makes NO_2_^−^ more easily adsorbed. In the presence of gold nanoparticles, energy barrier photogenerated electrons decrease the conduction band of NO_2_ molecules from the bottom. Therefore, NO_2_^−^ were adsorbed next to Au–C Schottky contact [55]. This behavior extends the width of the depletion region and increases the device resistance. In a dark nitrogen environment, the gas produced by carbyne nanocrystal has few electron-hole pairs, a very thin electron depletion layer on the surface and a high barrier height, resulting in high sensor resistance. When the sensor is exposed to light, carbyne nanocrystals produce electron-hole pairs, and the same electron depletion layer as in the N_2_ environment appears in the atmosphere in the dark. When exposed to NO_2_ gas, the photogenerated electrons react with NO_2_ molecules to form NO_2_, which is adsorbed on the carbyne nanocrystal surface to form a thick electron elimination layer, high barrier height and high sensor resistance, and the sensor’s gas sensitivity to NO_2_ is also further increased.

Zdenek Pytlicek studied anodized niobium oxide nanocrystalline films prepared by sputtering deposition [56]. The nanofilm consists of N_2_ layers that maintain upright N_2_O_5_ nanorods. It integrated each part into an advanced three-dimensional structure and multi-layer layout on a silicon wafer comprised of multiple micro-sensors, forming a top electrode and a multi-functional SiO_2_ sandwich by combining Pt/NiCr via high-temperature vacuum or air annealing, sputtering deposition, and stripping lithography. The proposed on-chip sensor solution enables sensitive fast and highly selective hydrogen detection. This thin film formation and chip manufacturing technology can enhance the gas sensing ability and efficiency of one-dimensional metal oxide nanomaterials.

Other advantages of this study are that the nanorods annealed in air, with lower oxygen vacancy concentrations, extending deeper within the rods and resulting in a flat-band state, thus leaving the oxide wholly depleted nanorods, which is associated with a significant increase in resistance after patching. The thickness of the depletion layer on the Schottky junction at the top of the rod also increases but remains unchanged in the vacuum annealed nanostructures between vacuum or air-annealed nanorods, respectively, with conductive channels inside or completely depleted H_2_ on the film resistance. It formed the track inside the rod, allowing a higher electronic conductivity than fully depleted nanorods in air. And the resistance decreases, resulting in a highly enhanced reaction to the H2. This will significantly enhance the sensing ability of the sensor on the high-capacity and low-power chip (In this part, through the analysis of the gas sensors composed of different kinds of one-dimensional nano materials, the principle, experimental steps, the differences from the traditional manufacturing method and the impact of this method are analyzed in detail. Finally, a detailed analysis, comparison and summary are provided for the researchers in the aspect of gas sensors composed of one-dimensional nano materials).

Hyoun Woo Kim and Sang Sub Kim synthesized SnO_2_-Cu_2_O C-S NWs and applied these to the detection of trace amounts of gases [57]. The resistance curves for the C-S NW sensors with different shell thicknesses were obtained upon exposure to 10 ppm C_7_H_8_, C_6_H_6_, and NO_2_. The sensor based on C-S NWs with a shell thickness of 30 nm exhibited the best response to reducing gases. The response of optimal gas sensor to 10 ppm C_7_H_8_, C_6_H_6_ gases was 11.7, 12.5 at 300 °C. In addition, the response and recovery times were almost 4 s for both gases. The presence of the Cu_2_O shell decreased the NO_2_-sensing response of the C-S NW sensors. In ambient air, the concentration of holes can be divided into three regions considering the vacuum case because of oxygen adsorption onto the Cu_2_O shell and development of the C-S heterojunctions. The HAL (p+) is created by the extraction of electrons from the valence band of Cu_2_O by chemisorbed oxygen species. At a specifific temperature, the intrinsic hole concentration layer (po) remains at the equilibrium hole concentration in Cu_2_O, and the hole-defificient layer (p−) results from an electrostatic response to the hole layer by the electrons in the n-p heterojunction. An increase in the concentration of holes is observed in air. When the sensor is exposed to the reducing gas, the resistance of the p-Cu_2_O shell layer increases.

The profile of hole concentration (blue line) in air shifts toward the red line, which supports a decrease in the concentration of holes in the “p” shell layer. Therefore, the detection capability of pure Cu_2_O NWs was inferior to that of the C-S NWs because of the weaker hole-accumulation layer. The degree to which the resistance of the p+ layer is modulated varies inversely with the shell thickness. As a result, a thicker shell experiences less resistance modulation because it is in a state of partial hole accumulation. Considering the fraction of shell layers in the overall volume of the n-p C–S NWs (which is comparable to shell thickness), the response affords a bell-shaped curve as a function of shell thickness. The extension of the p+ layer is constrained owing to the existence of the p−|n− interface, which acts as a blocking layer for the expansion of the p+ layer, resulting in a slight resistance modulation to oxidizing NO_2_ and low response to NO_2_.

## 4. Gas Sensors Based on Two Dimensional Nanomaterials

### 4.1. Gas Sensors Based on Graphene

Graphene gas sensors are widely used and play an essential role in industrial production and environmental monitoring. Due to its unique physical, chemical and mechanical properties, graphene has become a hot topic for many scholars. The gas sensor based on graphene material can reduce the working temperature, improve the recovery, and cooperate with other organic polymer materials to prepare the gas sensor used at room temperature. Graphene sensors show good gas sensitivity, which will have a wide range of applications in gas sensors.

Shirong Huang studied a flavin monoclinic sodium salt for ammonia gas sensing materials in a chemically resistive gas sensor [58]. The detailed characterization shows that the graphene sheets exhibit good structural quality and the optimized ammonia sensors exhibit excellent performance: ultra-low detection limits and excellent sensitivity to gases. Regarding the role of FMNS in graphene preparation and NH_3_ sensing, it studied the p-type doping of graphene-based sensing elements and the active adsorption site sensing for NH_3_ gases via full atomic molecular dynamics simulation. Using FMNS-like molecules to design high-sensitivity graphene-based NH_3_ gas sensors provides excellent sensing response and gas sensing capability.

A unique feature of the sensor is that the conductivity of the depleted graphene in graphene decreases, and the resistance increases when it rotates. Pure nitrogen, weak hydrogen bond breakage and NH_3_ molecular release bring electrons back when rinsing the sensor. Donated electrons move from the graphene return FMNS of the Fermi level to the valence state. It returned this to the lower sensor resistance and sensor recovery. Few electron energies from NH_3_ molecules are transferred to graphene, resulting in weak response sensors for graphene-based gases even when exposed to high ammonia concentrations. The specific modified hydrogen bond interaction of adsorbed NH_3_ molecules on FMNS is much more fragile, giving G-FMNS sensor orientation NH_3_ induction more than a covalent bond [59]. G-FMNS act as a graphene dispersion stabilizer as a P dopant element for graphene sensing and provide active adsorption sites for NH_3_ gas sensing, which constitutes an efficient graphene NH_3_ gas sensor and exhibits good gas sensing performance.

Muhanad. A. Ahmed studied the use of graphene nanosheets for ammonia gas detection [60]. The measurement is based on the measurement of electrical resistance before and after gas exposure. The resistance after gas exposure increased significantly. The graphene-based gas sensor was susceptible to NO_2_ and NH_3_. It was also prudent to other factors of organic compounds. Based on the adsorption of gas molecules, the resistance changed. Graphene responds quickly to gases using ultraviolet or heating sensors to provide sufficient energy for desorption from graphene sheets [61]. Graphene proved to be a P semiconductor, so after the weak adsorption of gas molecules adsorbed on the surface of graphene, the carrier was mainly hole hybridized with coupled electrons on the surface of graphene, which led to the change of graphene conductivity. Then, it showed the excellent sensing characteristics of specific gases and the sensing performance of the sensor.

Ravi Kumar investigated the preparation of functionalized graphene oxide (GO) films via chemical deposition [62]. It used the evaporation measurement technology of film resistance deposited by the thermal deposition process for aluminum contact. It characterized the functionalized graphene oxide by X-ray diffraction, Fourier transform infrared spectroscopy and Raman spectroscopy. The sensor response of ammonia concentration in a particular range was studied. MTA, in higher concentrations, had a higher response to ammonia. Increased ester generation reactions on the surface of the sensing film eventually led to interactions with NH_3_ gas molecules [63]. The selectivity of the sensor under various conditions was studied, and the sensor had strong selectivity to ammonia gas, which shows good sensing characteristics of ammonia gas.

The difference in this study is that carbon vacancies in oxygen species and FGO promote surface reactions with NH_3_, resulting in solid chemisorption and physical adsorption, and the adsorption and dissociation of ammonia on functionalized surfaces change structural and electronic properties. Functionalized GO and charge transfer occur from ammonia to functionalize the GO surface through the formation of surface hydrogen bonds. The presence of epoxy rings in the moving surface may break during functionalization. Moreover, the interaction process of the dissociation process of NH_3_ molecules also uses the sensing film, which it decomposed into ammonia and hydrogen [64]. Free NH_2_ and H molecules may react to obtain chemisorbed OH and NH_2_ molecules on available carbon and oxygen sites. The sensor will show excellent selectivity and gas sensitivity to ammonia.

Vu Van Cat studied GO nanosheets (GO-NSs) which were applied to a quartz crystal microbalance sensor for a mass-type poisonous gas sensor [65]. The sensor is a QCM working electrode surface prepared via spray of GO-NS suspension and it studied the concentration of toxic gases, including NO_2_, SO_2_, CO and NH_3_ via various methods of GO–NSs designed by the adsorbent which has good adsorption performance and becomes an efficient sensor for the detection of toxic gases.

In this study, the binding energy of NO_2_ molecules and GO is stronger than SO_2_ due to a reduction of oxygen-containing groups. The sensing result make known that the GO coated QCM sensor revealed a good sensitive factor (S-factor) with SO_2_ and NO_2_ gases, as exhibited in Figure 4. QCM sensors exhibit good response, long-term cycling stability and reproducibility. GO is graphene form containing oxygen functional groups with a negatively charged surface and surface large aspect ratio nanosheet structures. The sensitivity values of the QCM sensor SO_2_ of the go coating and the excellent sensitivity factor sensor of the NO_2_ gas are shifted from the frequency of the sensor to a specific concentration [66]. The working mechanism of the mass gas sensor is based on the adsorption/desorption layer of gas molecules on the sensor coated with QCM active electrodes. Based on the adsorption of dipole–dipole interaction, the groups on the GO surface and the polar test gas molecules pass through hydrogen bonds [67]. The S coefficient of the sensor depends on the adsorption capacity and the molecular weight of the adsorbate. The sensor has high sensitivity to NO_2_ gas and exists in the form of a dimer (N_2_O_4_) at a high density. The S factor of the sensor for NO_2_ gas is similar to that of SO_2_ gas and is higher than that for CO gas. The complex of the quality sensor GO exhibits a long response time for toxic gases such as sulfur dioxide and nitrogen dioxide, and the GO-coated QCM sensor exhibits extremely high gas sensitivity.

### 4.2. Gas Sensors Based on MXenes

With the rapid development of the chemical industry, the air pollution caused by organic waste gas emission has endangered people’s health and living environment. Toxic volatile gases, such as ammonia, amides, sulfur and nitrogen compounds are hazardous gases in the atmospheric environment. Therefore, it is essential to monitor the composition and concentration of harmful gases in the atmosphere in real-time and efficiently. A MXenes gas sensor is a sort of material which can detect the existence of specific types of gas and the change of concentration in a limited range in real-time. It shows high sensitivity and accuracy in gas detection and has great room for development in the future.

Shoumya Nandy Shuvo studied the gas sensing properties of volatile organic compounds by the lead of S doping Ti_3_C_2_TX–MXene, and it shows unique selectivity to toluene [68]. The study synthesized and characterized the sulfur-doped raw materials as electrode materials, and it subjected the as-prepared sensors to dynamic impedance tests at room temperature for the presence of volatile organic compounds with various functional groups. P-toluene has unique selectivity synthesized by undoped and doped Ti_3_C_2_TX-Mxene and obtains the sulfur-doped Ti_3_C_2_TX–MXene sensor, which embodies excellent long-term stability. The study revealed the effect of S doping on a volatile organic compound analyte sensing and obtained the efficient gas sensor for specific gas detection.

This study further understood the unique gas sensing behavior of these materials, as well as the selective doping of natural gas and natural gas Ti_3_C_2_TX-Mxene sensors. The main contribution near the Fermi level comes from the functionalization of Ti atoms, which leads to the modified electronic structure of the material. It shows that functional groups near the Fermi level promote functionalization of the atom-resolved band structure of the non-functionalized and functionalized titanium carbide MXenes. It used the binding energy of horizontal and vertical toluene and titanium carbide MXenes at four various positions for adsorption to form atoms that can find MXenes priority functional groups [69]. The flat binding energy and −S of toluene and titanium carbide MXenes as surface end groups at four various positions. The electron MXenes surface obtained from air enhances the gas response and exhibits an efficient gas sensor.

Danxi Yang determined that by using density functional theory and the first-principles method, SC_2_CO_2_ are very sensitive to NO molecules for their chemical interaction and large charge transfer, which will cause a change of current [70]. Also, it further enhanced the interaction by applying external strain, indicating the potential of SC_2_CO_2_ as a gas trapping agent material. Additionally, it studied the adsorption manganese doping of CO on SC_2_CO_2_. Due to the vital adsorption energy, this can gain a pronounced effect. The excellent sensing performance of SC_2_CO_2_ to CO in a gas sensor is reflected by the research.

This method differs from the traditional method. The interaction between molecules and gas sensing elements will detect the interaction between gas molecular materials and materials, resulting in a shift in material resistance. The adsorption of NH_3_ molecules on the Ti_2_CO_2_ is due to the change of charge carrier due to the transfer of charge between molecules based on the adsorption NH_3_ molecules and Ti_2_CO_2_. The current–voltage relationship and the adsorption energy of no molecules on the SC_2_CO_2_ is larger than that of NH_3_ molecules on the Ti_2_CO_2_, and the charge transferred to no molecules is more than that of the molecules on Ti_2_CO_2_. The sensitivity of SC_2_CO_2_ sensor is higher than that of Ti_2_CO_2_ sensor [71]. Adsorbed no molecules can be achieved by removing the applied biaxial strain. SC_2_CO_2_ shows good recognition and sensing ability of NO gas.

Shibin Sun studied the structure of W_18_O_49_/Ti_3_C_2_Tx composites based on the in situ growth of one-dimensional single crystals, using the solvothermal method to prepare W_18_O_49_ nanorods on the surface of 2D-Ti_3_C_2_Tx-Mxene sheets [72]. W_18_O_49_/Ti_3_C_2_Tx composites have a high response to low concentrations of acetone. W_18_O_49_/Ti_3_C_2_Tx composites showed significantly improved acetone sensing performance. It treated the uniform distribution of W_18_O_49_ on the Ti_3_C_2_Tx surface to remove the fluorinated group solvothermal process on the surface, as well as W_18_O_49_ NRs and Ti_3_C_2_Tx sheets. W_18_O_49_/Ti_3_C_2_Tx-Mxene composites show excellent sensing properties for acetone [73].

A unique feature of this method is that the W_18_O_49_/Ti_3_C_2_Tx sensor has highly enhanced acetone sensing capability. When the W_18_O_49_/Ti_3_C_2_Tx sensor is exposed to air, the surface of W_18_O_49_/Ti_3_C_2_Tx composite can adsorb air and O_2_ molecules. Oxygen molecules can capture electrons in oxide semiconductors and form oxygen species, reducing the electron concentration W_18_O_49_/Ti_3_C_2_Tx the resistance of the sensor. When exposed to acetone, acetone can react with ionic oxygen to release electrons [74]. The resistance of the W_18_O_49_/Ti_3_C_2_Tx sensor drops again. The oxygen vacancies of W_18_O_49_ can promote the adsorption of oxygen and acetone molecules in air and acetone, respectively, which helps to increase the resistance change of the W_18_O_49_/Ti_3_C_2_Tx composite material, resulting in a high response of the sensor. Acetone molecules are easily adsorbed on the Ti_3_C_2_Tx surface, and the adsorbed acetone can reduce the number of carriers on the Ti_3_C_2_Tx surface, resulting in an enhanced resistance change of the W_18_O_49_/Ti_3_C_2_Tx composite [75]. Moreover, a barrier layer can be formed between the Ti_3_C_2_Tx sheet and the W_18_O_49_ NRs, resulting in a decrease in resistance, and the sensor exhibits a greater resistance change; that is, a higher sensing response after contact with acetone [76] (This part discusses the most widely used two-dimensional nano materials in the field of gas sensors composed of nano materials, such as graphene, and performs a comprehensive and in-depth analysis and study of these materials, including their characteristics, specific action methods and their role in the development of sensors. This part also summarizes the gas sensor composed of two-dimensional nano materials, which is very comprehensive and perfect compared to the previous articles).

### 4.3. Gas Sensors Based on Other 2D Nanomaterials

Eunji Lee studied two-dimensional transition metal dihalides (TMDs) [77]. The inherent high specific surface area and their unique tunable band semiconductor characteristic gap make them attractive for sensing applications. In combination with two-dimensional nanomaterials, metal oxides are incorporated into two-dimensional transition metal dihalides, and the synergistic effect is used to improve the gas-sensing performance of these materials. The sensing mechanism and the synergistic effect of hybridization of 2D-TMDs and metal oxides were studied. 2D-tmd and its metal oxide hybrid material TMD have excellent capabilities in gas sensing.

The difference in this way is that the charge transfer process is between the gas molecule and the surface of the sensing material. 2D materials act as charge acceptors or donors and produce changes in electrical resistance. Exposed to reactive gases, gas molecules are electrostatically adsorbed on the surface of two-dimensional materials. The directional transfer of electron charges depends on the type of reductive or oxidizing reaction gas [78]. After resistance due to the desorption of gas molecules, the sensing material returns to its original value. Under the exposure of reducing gas, the resistance of the sensing material increases. At the working temperature, the target gas is introduced, and the electrons in the reducing gas are transferred to the conduction band of the metal oxide, resulting in a decrease in the resistance of the sensor. However, NH_3_ acts as a reducing gas due to its lone electrons, and due to the leading role of oxygen ions, it exhibits the gas sensitivity and sensing performance of the gas sensor [79].

Bingshan Wang studied the synthesis of two-dimensional mesoporous ZnSnO_3_ nanomaterials (TMZNS) by employing a template-free hydrothermal method [80]. The product was characterized by differential thermal analysis by TEM and SEM. The high-purity TMZNS was prepared by the hydrothermal method using a mixture of Zn_5_(OH)_6_(CO_3_)_2_ and ZnSnO_3_ as the precursor. The two-dimensional mesoporous structure was studied through the process and mechanism of quasi-crystal growth. The gas-sensitivity characteristics of TMZNS were studied and characterized, and sensors based on TMZNS had excellent gas-sensitivity performance. When exposed to formaldehyde vapor, the sensor exhibited good gas sensitivity and sensing ability to formaldehyde gas [81].

In this research, the special feature is the surface of conductive material that occurs in the air, including a series of adsorption–oxidation–desorption. When TMZNS is in the air, a large number of oxygen molecules are adsorbed on the surface of the TMZNS sensor layer. The electrons on the surface of the TMZNS sensing layer are captured and adsorbed by a large amount of oxygen, which causes the trans-generation of oxygen species to oxygen ions to form a layer where the operating temperature and electrons are depleted. The barrier and resistance are relatively high. There is exposure to formaldehyde gas, formaldehyde gas interacts with the surface and it releases oxygen and electrons back into the air sensing materials. Chemically adsorbed oxygen and porous nanopores can improve sensing performance. A large specific surface area provides larger contact area and voids, more chemically active sites and less band gap energy helps the conduction band of the research ZnSnO_3_ for the adsorption and capture of electrons on the surface [82]. Unique perforated nanopores are beneficial for adsorption to shorten the gas diffusion path. Therefore, TMZNS becomes an efficient gas sensor for HCHO detection.

Hanie Hashtroudi studied the gas sensing properties of two-dimensional hybrid nanomaterial conductance devices containing layered transition metals at room temperature and the effect of two-dimensional nanomaterials hybridization on the gas sensing properties [83]. By adding metals or polymers to the surface location, or by combining two or more different materials or by developing heterojunction layers, the sensing performance is improved. The gas is adsorbed on the active surface, which leads to a change in resistance, which in turn greatly improves its gas sensing capability [84].

The unique feature of this method is that the hybridization of various kinds of nanomaterials improves the sensitivity and acceleration response and recovery of the sensor. The interaction between gas molecules introduces new physical and electrical properties, thus improving the gas sensing performance. Surface engineering induced changes in the thickness and number of reaction sites for gas molecular interactions. The conductivity and charge transfer of the hybrid sensing, and the Fermi properties of the electrical and heterojunction layers at their interfaces change the directional energy differences of the current-carrying flow. Figure 5 demonstrates the dynamic response, sensitivity, and the selectivity of these two sensors for the NO_2_ sensing at RT. Surface functionalization and unique treatment improve selectivity, sensitivity and specific sensing parameters and significantly improve the gas sensing characteristics and sensing performance of the sensor to gas in a natural environment [85].

## 5. Conclusions

The advantages of various dimension nanomaterials constitute gas sensors, which may become the development direction of a new generation of gas sensors due to their high sensing efficiency, good gas detection ability and high sensitivity. In this review, the development of gas sensors composed of nanomaterials is discussed from various dimensions of nanomaterials, sensing effect and sensing mechanism. At the same time, it reviewed the gas sensors composed of other dimension nanomaterials from the methods used, the characteristics of various categories and the advantages and contributions of each study. At present, most of the gas sensors have entered the nanometer field, which has a significant advantage over the traditional sensors, and will provide a new way to solve the problem of gas sensor sensing efficiency and sensing ability in gas detection. Future gas sensors composed of nanomaterials will have multiple functional characteristics, such as high gas sensing efficiency, high sensing sensitivity and a strong ability to detect specific gases, providing better development in the sensor field.

## Figures and Tables

**Figure 1 micromachines-13-00919-f001:**
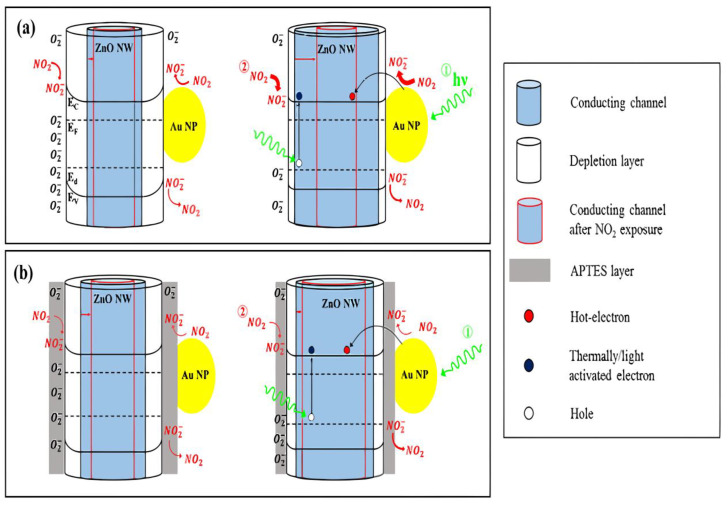
Schematic mechanisms for the enhanced NO_2_ gas response of the Au nanoparticles-decorated ZnO nanowires, (**a**) without (Au–ZnO) and (**b**) with a (3-aminopropyl) triethoxysilane layer (Au–ZnO/APTES), in the dark (**left**) and under green illumination (**right**). The curved red arrows indicate the NO_2_ gas adsorption and desorption processes, whose acceleration and diminishment are represented by the arrow thickness. The solid and dashed lines denote, respectively, the intrinsic conduction (EC) and valence (EV) bands and the Fermi (EF) and defect (Ed) levels. Reprinted with permission from Ref. [20].

**Figure 2 micromachines-13-00919-f002:**
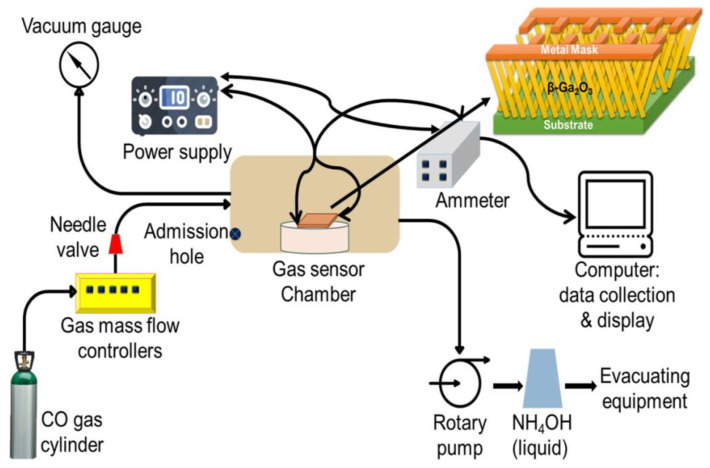
An architecture diagram of a CO gas sensing measuring unit. Reprinted with permission from Ref. [31].

**Figure 3 micromachines-13-00919-f003:**
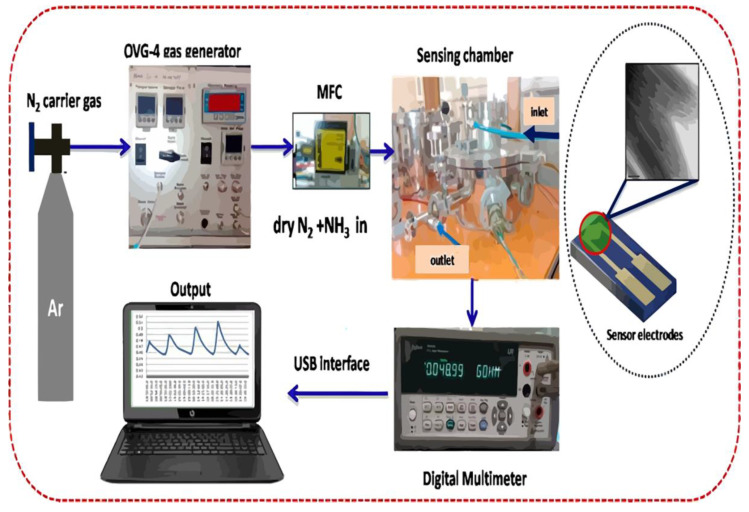
A schematic illustration of the experimental setup used for NH_3_ gas sensing measurements.

**Figure 4 micromachines-13-00919-f004:**
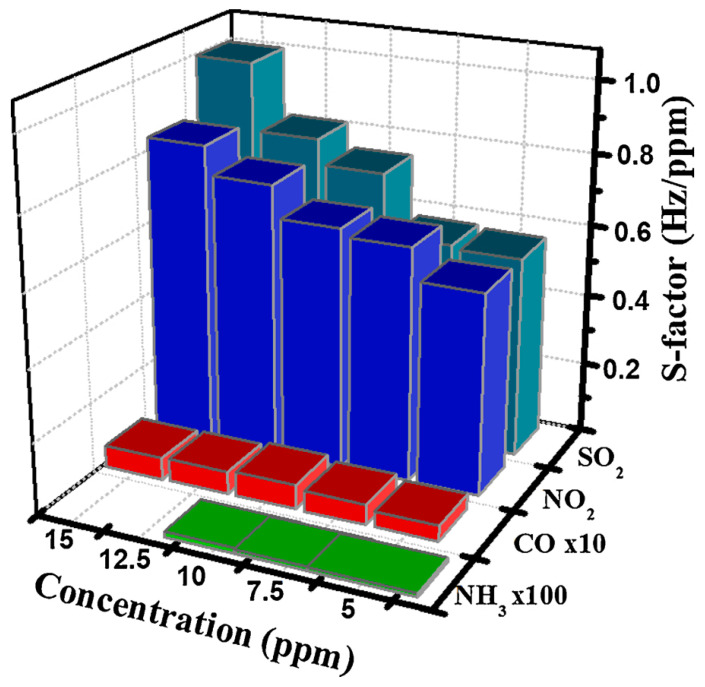
The sensitive factors (S-factor) of SO_2_, NO_2_, CO and NH_3_ at different concentrations.

**Figure 5 micromachines-13-00919-f005:**
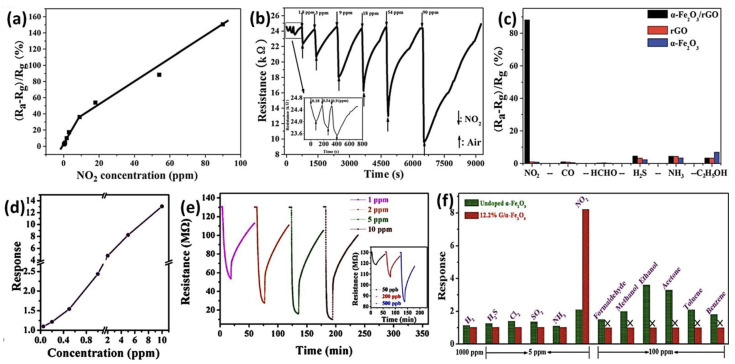
(**a**) The response of α-Fe_2_O_3_/rGO nanocomposite as a function of NO_2_ concentration at RT, (**b**) dynamic responses of α-Fe_2_O_3_/rGO nanocomposite towards different NO_2_ concentrations, and (**c**) selectivity of α-Fe_2_O_3_/rGO nanocomposite, rGO, and α-Fe_2_O_3_ to different gases at RT. Adapted with permission.106 (**d**) The response of 12.2% α-Fe_2_O_3_/rGO as a function of NO_2_ concentration at RT, (**e**) dynamic response of 12.2%α-Fe_2_O_3_/rGO at different NO_2_ concentrations, and (**f**) selectivity of 12.2% α-Fe_2_O_3_/rGO and α-Fe_2_O_3_ towards various gases. Adapted with permission.

**Table 1 micromachines-13-00919-t001:** A brief summary of nanomaterials with different dimensions.

Dimension	Typical Materials	Characteristics	Work of Current Research
Zero dimension	Nanoclusters, Metal Nanoparticles	Photostability, Biocompatibility Light Induced Flourescence, Sensing Performance	Fabricate dual-terminal gas/TiO_2_ sensor devices using a top-down approach. Gas nanosensor system combining silver itnanoclusters with phosphorene. Synthesize heterostructured nanomaterials composed of p-type TeO_2_ NWs and discrete n-type Use ZnO nanoclusters to detect NO_2_ gas molecules. The effects of gold nanoparticles on the performance of CO gas sensing sensors.
One dimension	Nanowires, Carbon Nanotubes, Nanofibers, Nanorods	Heat transfer, Photocatalysis, Small size effect, Conductivity	Synthesize SnO_2_ nanowires modified by Pd nanoparticles to prepare highly selective and sensitive nanowire hydrogen sensors. Studied the room temperature CO gas sensor of pure nanowires and gold-modified nanowires using multi-network arrays and single nanowire devices. Study SnO_2_ nanowire hall-effect gas sensor for hydrogen detection. Study the chemical detection of toxic gases by gas sensors based on single-walled carbon nanotubes to detect NO_2_ gases with higher sensitivity.
Two dimension	Graphene, MXENES	High hardness Thermal performance, Excellent sensing performance, Flexibility Optical performance	Study GO nanosheets were applied to a quartz crystal microbalance sensor for a mass-type poisonous gas sensor. Study a flavin monoclinic sodium salt for ammonia gas sensing materials in a chemically resistive gas sensor. Study the use of graphene nanosheets for ammonia gas detection. Study the gas sensing properties of volatile organic compounds by the lead of S doping Ti_3_C_2_TX-Mxene, and it shows unique selectivity to toluene.

## Data Availability

Not applicable.

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
