# Peer review of "Recent Progress in Gas Sensor Based on Nanomaterials"

_micromachines, 2022, doi:10.3390/mi13060919_

Round 1
Reviewer 1 Report
I would like to congratulate the authors on putting together a fine review paper. It is broad but still focused on the subject matter.
There are some issues that need to be addressed.
First, I believe the table on the sensors should be moved to the end of the introduction.
Next, in section 2.1, line 87 has a sentence that starts "rGO-CDs the composite..." Please rework that sentence to be much more easily read.
Third, there is a large sentence run on sentence from line 108 to line 11. Please break up for be more understandable.
Fourth, In Figure 2, please move the close up of the sensor to below the Gas sensor chamber. The arrow that directs the reader from the small orange square to the close up is distracting and crosses the Ammeter/electrical lines. The rest of the figure is easy to understand.
Finally, while I understand there are limitations, I believe that there should be more figures/diagrams to better illustrate the some of the sensors and more importantly their responses.
Author Response
Dear Ph. D. Reviewer,
Thank you very much for your attention to our paper. We have revised the manuscript in detail according to your suggestion. All changes to the original manuscript are marked by the color of red. It is sincerely hoped that this manuscript on the subject of “Nanostructure based Sensors for Gas sensing: From Devices to Systems” in Micromachines will finally be accepted and published. Thank you very much for all your help and look forward to hearing from you soon.
Best regards.
Sincerely yours,
Dr. Ke Xu
Prof. Ke Xu
Please find the following response to the comments of reviewer and editor.
Response to the referee and editor’s comments.
Reviewer 1: I would like to congratulate the authors on putting together a fine review paper. It is broad but still focused on the subject matter. There are some issues that need to be addressed.
Response to Reviewer 1 Comments
Point 1: I believe the table on the sensors should be moved to the end of the introduction.
Response 1: Thanks for the referee’s kind advice. We have moved the table to the end of the introduction in page 3.
Point 2: In section 2.1, line 87 has a sentence that starts "rGO-CDs the composite..." Please rework that sentence to be much more easily read.
Response 2: Thanks for the reviewer’s correcting. We are so sorry for these problems in the review. And we have made careful revisions in accordance with the requirements in page 4.
“ This process results in the formation of a depletion layer on the rGO-CDs surface. rGO-CDs the composite contact gas with NO2, there are grain boundaries at the same time in many active sites, it used the whole carbon nanoscale for the selectivity of NO2 heterojunction molecules. ”
is changed to
“ This process results in the formation of a depletion layer on the rGO-CDs surface. There are grain boundaries at the same time in many active sites in rGO-CDs and the composite contact gas with NO2, it used the whole carbon nanoscale for the selectivity of NO2 heterojunction molecules.”
Point 3: There is a large sentence run on sentence from line 108 to line 111. Please break up for be more understandable.
Response 3: Thanks for the reviewer’s kind advice. We are so sorry for the sentence problem in the review, and the parts that have sentence problems have been revised in page 4.
“And after combining with the carbon dots, the successful electron transfer, the formation of the heterostructure of In2O3 and the carbon dots; the shielding effect due to the introduction of carbon dots, and the increase in the percentage of oxygen vacancy and surface adsorption oxygen of the introduction of carbon dots, the higher the carbon dots several reasons, such as active surface effect, show excellent sensing performance for NO2 and other gases.”
Is changed to
“And after combining with the carbon dots, the successful electron transfer, the formation of the heterostructure of In2O3 and the carbon dots. The shielding effect due to the introduction of carbon dots, and the increase surface adsorption oxygen of the introduction of carbon dots. Carbon dots has several reasons, such as active surface effect, show excellent sensing performance for NO2 and other gases.”
Point 4: In Figure 2, please move the close up of the sensor to below the Gas sensor chamber. The arrow that directs the reader from the small orange square to the close up is distracting and crosses the Ammeter/electrical lines. The rest of the figure is easy to understand.
Response 4: Thanks for the reviewer’s advice. We have followed the reviewer’s suggestion and changed the figure in page 14.
Point 5: I understand there are limitations, I believe that there should be more figures/diagrams to better illustrate the some of the sensors and more importantly their responses.
Response 5: Thanks for the reviewer’s kind advice. We are so sorry for the problem in the review, and we add new figure in page 27. Thanks again for your kind suggestions.

Reviewer 2 Report
Based on the review paper submitted to the journal, my comments are as follows:
1- the number of Figure can be increased. More figures are necessary for a review paper in this level. In particular, more emphasis should be related to sensing mechanism figures.
2- Just reporting of a paper is not enough and the authors should pay attention to the sensing mechanism and sensing enhancement reasons.
3- More gas sensors such as flexible sensors, electron beam irradiated gas sensors, ion- implanted gas sensors, MOF has sensors and so on should be included.
4- The works of professor Hyoun Woo Kim and Sang Sub Kim are related to core- shell gas sensors and other one dimension gas sensors and therefore they are recommended to be included in this review.
And more logical connection should be between different sentences and paragraphs. Therefore whole of manuscript should be re-checek.
Author Response
Dear Ph. D. Reviewer,
Thank you very much for your attention to our paper. We have revised the manuscript in detail according to your suggestion. All changes to the original manuscript are marked by the color of red. It is sincerely hoped that this manuscript on the subject of “Nanostructure based Sensors for Gas sensing: From Devices to Systems” in Micromachines will finally be accepted and published. Thank you very much for all your help and look forward to hearing from you soon.
Best regards.
Sincerely yours,
Dr. Ke Xu
Please find the following Response to the comments of referees:
Response to the referee’s comments.
Reviewer 2: Based on the review paper submitted to the journal, my comments are as follows.
Response to Reviewer 2 Comments
Point 1: The number of Figure can be increased. More figures are necessary for a review paper in this level. In particular, more emphasis should be related to sensing mechanism figures.
Response 1: Thank you very much for your suggestion. According to your suggestion, we added the figure in page 27.
Point 2: Just reporting of a paper is not enough and the authors should pay attention to the sensing mechanism and sensing enhancement reasons.
Response 2: Thank you very much for your suggestions. According to your suggestions, we have added the sensing mechanism and sensing enhancement reasons in page 7, 11, 20 and 27.
Point 3: More gas sensors such as flexible sensors, electron beam irradiated gas sensors, ion-implanted gas sensors, MOF has sensors and so on should be included.
Response 3: Thank you very much for your suggestion. According to your suggestion, we added the corresponding content in page 26.
Point 4: The works of professor Hyoun Woo Kim and Sang Sub Kim are related to core- shell gas sensors and other one dimension gas sensors and therefore they are recommended to be included in this review.
Response 4: Thank you very much for your suggestion. According to your suggestion, we added the works of professor Hyoun Woo Kim and Sang Sub Kim are related to core- shell gas sensors in page 22.
Point 5: More logical connection should be between different sentences and paragraphs.
Response 5: Thanks for the reviewer’s kind advice. We are very honored to be recognized by you. We do hope this review will promote more research on gas sensor based on nanomaterials and accelerate its development and applications. Through reading this review, readers can ask for the latest comprehensive information, understand the new progress, problems, and efforts direction of the discipline about gas sensor based on nanomaterials. We are so sorry that there are some problems in manuscript. Now, this manuscript has made more logic connection between different sentence and paragraphs. Thanks again for your valuable suggestion.

Round 2
Reviewer 2 Report
The revised version has many shortages.
1- For example, the authors have added some results but the number of references is not changed. In Page 21, the authors should add the reference.
2- In Table 1, the authors have not added nanofibers and nanorods into one dimension category.
3- In section 2.1, the first paragraph has no reference. Thus, please add references to add parts that need reference.
4- In Page 4, you have mentioned Cheng Ming, but the relevant reference is missing.
5- Overall all references should be double checked.
6- In section 2.3 only Gold is discussed. Please add the papers related to other noble metals such as Pt, Pd and Ag.
7- Also, the first paragraph section 3.1 has no references.
8- Overall, please check it with outmost care again... there are many shortages should need to be fixed.
Author Response
Dear Ph. D. Reviewer,
Thank you very much for your attention and the reviewer’s evaluation and comments on our paper. Those comments are all valuable and very helpful for revising and improving our paper. We have studied the comments carefully and have made correction which we hope to meet with approval. All changes to the original manuscript are marked by the color of red. It is sincerely hoped that this manuscript on the subject of “Nanostructure based Sensors for Gas sensing: From Devices to Systems” in Micromachines will finally be accepted and published. Thank you very much for all your help and look forward to hearing from you soon.
Best regards.
Sincerely yours,
Ph. D. Ke Xu
Please find the following Response to the comments of referees:
Response to the referee’s comments.
Reviewer: Based on the review paper submitted to the journal, my comments are as follows.
Response to Reviewer Comments
Point 1: For example, the authors have added some results but the number of references is not changed. In Page 21, the authors should add the reference.
Response 1: Thank you very much for your suggestion. According to your suggestion, we added the reference in page 21, and marked them in the color of red.
Point 2: In Table 1, the authors have not added nanofibers and nanorods into one dimension category.
Response 2: Thank you very much for your suggestions. According to your suggestions, we have added nanofibers and nanorods into one dimensional category in table 1 inpage 2.
Point 3: In section 2.1, the first paragraph has no reference. Thus, please add references to add parts that need reference.
Response 3: Thank you very much for your suggestion. According to your suggestion, we added the references in page 29, and marked it in the color of red.
Point 4: In Page 4, you have mentioned Cheng Ming, but the relevant reference is missing.
Response 4: Thank you very much for your suggestion. We put the reference in the first position in page 29.
Point 5: Overall all references should be double checked.
Response 5: Thanks for the reviewer’s kind advice. We have made a detailed examination of the references, and we marked the wrong reference in the color of red.
Point 6: In section 2.3 only Gold is discussed. Please add the papers related to other noble metals such as Pt, Pd and Ag.
Response 6: Thanks for the reviewer’s kind advice. We have added the papers related to other noble metals in page 6.
Point 7: Also, the first paragraph section 3.1 has no references.
Response 7: Thanks for the reviewer’s kind advice. We have added reference in page 30, and marked it in the color of red.
Point 8: Overall, please check it with outmost care again.
Response 8: Thanks for the reviewer’s kind advice. We are very honored to be recognized by you. We do hope this review will promote more research on gas sensor based on nanomaterials and accelerate its development and applications. Through reading this review, readers can ask for the latest comprehensive information, understand the new progress, problems, and efforts direction of the discipline about gas sensor based on nanomaterials. We are so sorry that there are some problems in manuscript. Now, this manuscript has made a detailed examination. Thanks again for your valuable suggestion.

Round 3
Reviewer 2 Report
Based on the revised version, it can be accepted now...